# Current Status of Omics Studies Elucidating the Features of Reproductive Biology in Blood-Feeding Insects

**DOI:** 10.3390/insects14100802

**Published:** 2023-10-06

**Authors:** Aditi Kulkarni, Frida M. Delgadillo, Sharan Gayathrinathan, Brian I. Grajeda, Sourav Roy

**Affiliations:** 1Department of Biological Sciences, University of Texas at El Paso, El Paso, TX 79968, USA; aditi.kulkarni@seattlechildrens.org (A.K.); fmdelgadillo@miners.utep.edu (F.M.D.); sgayathrina@miners.utep.edu (S.G.); bigrajeda@utep.edu (B.I.G.); 2Border Biomedical Research Center, University of Texas at El Paso, El Paso, TX 79968, USA; 3Environmental Science and Engineering Ph.D. Program, University of Texas at El Paso, El Paso, TX 79968, USA; 4Biosciences Ph.D. Program, University of Texas at El Paso, El Paso, TX 79968, USA

**Keywords:** insect vectors, insect bioinformatics, mosquitoes, kissing bugs, tsetse flies

## Abstract

**Simple Summary:**

Insect vectors are responsible for transmitting a range of diseases, leading to significant mortality rates annually. Their behavior and physiology can undergo shifts due to complex molecular interactions during mating and feeding. In this review, we provide an exhaustive overview of the current “omics” knowledge—spanning genomics, transcriptomics, proteomics, and metabolomics—across various vector species. We highlight potential molecular targets for vector control and outline the advancements and gaps in our understanding, which could pave the way for innovative and effective strategies to curb disease transmission.

**Abstract:**

Female insects belonging to the genera *Anopheles, Aedes, Glossina,* and *Rhodnius* account for the majority of global vector-borne disease mortality. In response to mating, these female insects undergo several molecular, physiological, and behavioral changes. Studying the dynamic post-mating molecular responses in these insects that transmit human diseases can lead to the identification of potential targets for the development of novel vector control methods. With the continued advancements in bioinformatics tools, we now have the capability to delve into various physiological processes in these insects. Here, we discuss the availability of multiple datasets describing the reproductive physiology of the common blood-feeding insects at the molecular level. Additionally, we compare the male-derived triggers transferred during mating to females, examining both shared and species-specific factors. These triggers initiate post-mating genetic responses in female vectors, affecting not only their reproductive success but also disease transmission.

## 1. Introduction

Humans are infected with various infectious agents throughout their lifetime, and about 17% of these infectious diseases are transmitted by insect vectors [1]. Annually, approximately 700,000 deaths are reported from diseases such as malaria, dengue, chikungunya, zika, human African trypanosomiasis, Chagas disease, leishmaniasis, yellow fever, and Japanese encephalitis [1,2,3,4]. Malaria and dengue are the most common vector-borne diseases affecting millions of individuals across the globe, with the highest burden in tropical and sub-tropical regions [1,5]. While sleeping sickness (African trypanosomiasis) and Chagas disease (American trypanosomiasis) have been subjects of study for over a century, they continue to afflict an estimated 6–7 million people each year [6,7]. 

All of these aforementioned diseases are categorized as lethal due to the harm they cause on a global scale. American trypanosomiasis has an average incidence of 30,000 new cases and 12,000 deaths annually, with around 9,000 newborns being infected during gestation [8]. African trypanosomiasis, by comparison, has shown a significant reduction in cases due to medical and government interventions, but about 1000 cases are still reported annually, with 3 million at moderate risk as of 2018 [7]. Chikungunya virus, while rarely fatal (less than 1 per 1000 cases) [9], can cause arduous prolonged joint pain in older patients [10,11,12,13]. In 2021, there were around 247 million cases of malaria worldwide and an estimated 619,000 deaths [14]. Unfortunately, children under the age of 5 account for 80% of these deaths in African regions, a disproportionately high number compared to the global average [15]. 

It is estimated that 109,000 severe infections of yellow fever occurred in Africa and South America in 2018 alone, leading to approximately 51,000 deaths [16]. As of 2023 Yellow fever continues to be endemic in 47 countries, 13 of which are found in South America and 34 in Africa [17]. In 2016, Zika was declared to be a Public Health Emergency of International Concern (PHEIC), as it caused infants to be born with microcephaly and other congenital malformations [18,19]. Though there has been a global decline, transmission persists in several countries in the Americas and other endemic regions [20]. Countries in the Northern Hemisphere are also at risk, as the parasitic disease Leishmaniasis claims over 50,000 lives annually and is an emerging health risk in Mediterranean, North American and other European countries [2].

All of these diseases depend on insect vectors for transmission, with each vector species harboring its unique array of viruses and parasites, coupled with distinct behavior and feeding patterns. For instance, *Anopheles gambiae*, known for its transmission of malaria and dengue, displays both endophilic and endophagic behaviors. They rest and feed indoors [21,22] and are also known to feed during the night [22]. They prefer human blood over other hosts [23], producing a close relationship with humans [24,25]. *Aedes aegypti*, a transmitter of dengue, chikungunya, zika and yellow fever, has been found to rest both in- and outdoors [25,26,27] and is bimodal, generally feeding in the early morning and late afternoon [28,29]. Their feeding periods may allow the vectors to avoid long-lasting insecticidal nets (LLINs) since they generally feed outside of when the host is asleep [25]. *Glossina morsitans,* commonly known as the tsetse fly, transmits sleeping sickness or African trypanosomiasis [30]. They have a wide geographical distribution across 24 countries of west and central Africa and 13 countries of eastern and southern Africa [7]; however, less is known about their resting and feeding behaviors. *Rhodnius prolixus*, or triatomines, transmit Chagas disease and are prevalent in the Americas, particularly in Brazil and the USA [31,32]. Triatomines are nocturnal, feeding on hosts while they sleep and hiding during the daytime in indoor cracks and crevices [33,34]. *Phlebotomus papatasi*, also known as sand flies, are known to harbor leishmania as well as Bartonella and arboviruses belonging to the Phlebovirus genus [35]. These flies are most active during twilight and late-night hours and are found in various regions across Asia, the Middle East, Africa, Southern Europe, Mexico, Central America, and South America [36].

Despite the evident need, there is no effective and licensed vaccine for malaria, sleeping sickness, or Chagas disease [6,7,37]. On the other hand, the FDA has approved a dengue vaccine [38], but it is recommended only for individuals with confirmed prior dengue virus infection [39,40]. Yellow fever vaccines are also available, but intervention strategies still need optimization [16]. Control measures still depend largely upon early case detection, treatment, and vector control. Vector control strategies deployed globally include habitat reduction [41,42], structural barriers [41,42], and chemical [41,43,44] and biological control methods [45,46,47,48,49]. Even though these are successful, additional efforts are required to identify new tools for vector control to overcome the difficulties associated with the current strategies like insecticide resistance or sustainability in the environment. Understanding the basic biology and molecular responses associated with various physiological changes occurring in female disease vectors could help us identify unique targets to come up with new and improved vector control methods, which may be specific or shared among co-existing species. The advancement of omics technologies, including genomics, transcriptomics, proteomics, and metabolomics, in the past few years has led to the characterization of the molecular, cellular, and functional biology of various insect vectors.

In this review, we present the current knowledge about the genomic, transcriptomic, proteomic, and metabolomic studies describing the reproductive biology of four blood-feeding female insect vectors, *Anopheles gambiae*, *Aedes aegypti*, *Glossina morsitans*, and *Rhodnius prolixus*. Further, we compare the recent findings concerning genes responsible for male seminal fluid synthesis and transfer of male-derived triggers in mating and post-mating genetic responses in female insects [50] to help us understand the reproductive components that are still unexplored in some insect vectors, and how building this knowledge gap could help us identify potential targets for vector control. Information obtained from these evolutionary studies would provide us with strategies for targeting this complex physiological process to prevent successful mating or reproduction in several insect vectors, which may eventually lead to a reduction in the transmission of vector-borne diseases globally.

## 2. Genomic Studies

Developing specialized and efficient vector control measures requires a solid understanding of the reproductive biology of disease vectors. The genetic mechanisms underpinning reproductive behaviors may now be examined, providing a more detailed understanding of vector ecology. This is made possible by advancements in genomic analysis. 

Successful reproduction does not solely depend on the female’s efforts. The male’s accessory gland (MAG) secretions are not only crucial for efficient sperm transfer and subsequent storage but also play a pivotal role in modulating female behavior and physiology post-mating [51,52]. Seminal fluid proteins (SFP) have been seen to induce specific behaviors in females like reduced sexual receptivity and stimulated oviposition, serving as potential targets for innovative mosquito control strategies aimed at altering female behavior or reproductive success [53] (Figure 1). Omics-based approaches can unravel the specific protein patterns and gene expressions that underlie the functionality of MAG secretions and seminal fluid proteins. MAG genes and the accessory gland proteins (Acps) they encode play a critical role in controlling post-mating behavioral changes in females. Preliminary studies in *An. gambiae* have identified 46 MAG genes that encode for the respective Acps [54], which are responsible for post-mating behavioral changes in females [55,56,57] (Figure 1;Table 1). However, the orthologs for the genes encoding for the putative male reproductive gland proteins (mRGPs) identified in *Ae. aegypti* could not be identified in *Culex*, other *Anopheles*, or *Drosophila* species [56] (Table 1). More than 50% of the Acps identified in *An. gambiae* represented a novel lineage when compared to *Drosophila* [57], suggesting that these proteins are drastically evolving and contributing to specific and relevant reproductive functions in these disease vectors (Table 1). Recent advancements in sequencing and the publication of the genomes of *Anopheles* species [58], *Ae. aegypti* [59,60], *Gl. morsitans* [61], and *R. prolixus* [62] have provided exceptional opportunities to understand the important aspects of the reproductive biology of these vector species. 

In the mosquito world, monogamy is mostly a female trait; most female mosquitoes are believed to mate just once in their lifetime. Despite this, they have the ability to produce eggs in each subsequent gonotrophic cycle following a blood meal. On the other hand, male mosquitoes are known for their propensity to re-mate [63]. In *Ae. aegypti* [64] or *An. gambiae* [65], the refractoriness to multiple copulations is largely a result of SFPs that are transferred during mating (Figure 1; Table 1). Specifically, in *An. gambiae*, the inefficient transfer of the hormone 20-hydroxyecdysone (20E) has been identified as a key factor that reinforces this monogamous behavior in females, making it a crucial post-mating response [65] (Table 1). The molecular divergence explained by comparative genomics between vector species can be used for designing new vector control strategies. Another gene, Or, which was originally found in *An. gambiae* to be associated with sperm activation [66], was also later identified in *Stomoxys calcitrans* (stable fly), showing 17 Or genes relevant for reproduction in mated males [67] (Figure 1; Table 1). *S. calcitrans* has recently had many current and novel transcript reproductive enrichments identified.

During their life span, female insect vectors undergo various physiological, behavioral, and transcriptional changes. The females mate only once or multiple times, depending on the species [50,63,68,69,70,71], to acquire sperm and seminal fluids from the male population. Post-mating, blood feeding is necessary for these females to obtain nutrients required for energy and reproduction. This requirement has been highly exploited by various parasites and viruses, making the reproductive process very complex, and significant for vector competence. In species that reproduce by laying eggs, such as certain blood-feeding insects, the yolk serves as a carefully arranged nutritional reservoir allocated by the female for embryonic growth. Upon hematophagy, the insect vectors initiate a cascade of physiologic events geared toward oogenesis. Oogenesis is the development and differentiation of the ovum, which allows for the development of an embryo once fertilized [72]. Blood-derived amino acids facilitate yolk protein precursor (YPP) synthesis via proteolytic activity, which provides the initial forms of proteins that are eventually incorporated into the yolk of insect eggs, while vitellogenesis is marked by upregulated Vg synthesis and transport to oocytes via the hemolymph, a fluid found in most invertebrates that is equivalent to blood. Concurrently, lipid reserves are mobilized and also used for the synergic shift toward egg maturation. Collectively, these processes culminate in oocyte maturation, prepping the insect for subsequent embryogenesis [73,74,75]. The oocyte, representing the female’s haploid gamete destined to develop into the embryo following fertilization, plays a pivotal role in accumulating these crucial nutrients. This accumulation process takes place during the vitellogenic stage, characterized by substantial alterations in the oocyte’s size and morphology primarily attributed to nutrient uptake [76].

The efforts to sequence 16 *Anopheles* genomes laid the foundation for using comparative genomics to define nucleotide-level evolution across genera [58]. Initial work with *An. gambiae* had demonstrated that the crosslinking activity of a MAG-specific transglutaminase enzyme, AgTG3, is required for the formation of a mating plug [77,78] (Figure 1; Table 1). It was believed that the purpose of the plug was to completely occlude the female reproductive tract, ensuring the reproductive success of the *Anopheles* males [77,78]. However, in a follow-up phylogenetic analysis, AgTG3 was observed to be highly evolved as compared to the other two transglutaminase enzymes present across the 16 *Anopheles* species [58], which could represent the extensive divergence in mating plug phenotypes [58]. Further, in a phenotypic study, it was established that *An. gambiae*, *An. arabiensis*, *An. funestus* and *An. stephensi* had a completely coagulated mating plug with higher levels of 20E as compared to *An. atroparvus, An. dirus, An. farauti* and *An. sinensis*, which showed to have an intermediate level of coagulation and 20E synthesis. These phenotypes were drastically different from the New World species, *An. albimanus*. In the males of this species, plug formation and 20E synthesis were completely absent [79].

Besides 20E synthesis and mating plug formation, an additional study in *An***.**
*gambiae* revealed that the yellow protein gene (*yellow-g*) is crucial for female reproduction [80] (Figure 1). Coupled with the availability of the *An. stephensi* reference-grade genome assembly, complete transcript annotations were produced, revealing three other yellow protein gene family members (*yellow-b*, *yellow-e*, and *yellow*) that displayed higher transcript levels post-blood meal (PBM) [81].The four yellow genes in *An. stephensi* have PBM upregulation patterns that are consistent with roles in female reproduction, though *yellow-b*, *yellow-e*, and *yellow* are likely to be needed for a longer period than *yellow-g* [81] (Table 1). However, this clear evolutionary characterization is still absent in other insects, including *Ae. aegypti*. In this vector species, a gene that encodes for a neuropeptide, HP1, has been characterized so far, to induce female refractoriness to re-mating, which is a result of a gene duplication event of a neuropeptide (F gene) in an ancestor *Aedes* species [82] (Table 1). With the publication of *Ae. albopictus* genome [83] and improvements to the assembly of the *Ae. aegypti* genome [60], it could be taken up as an excellent platform for conducting similar comparative analyses of the genes that code for SFPs that, upon delivery, induce several behavioral and physiological changes in female *Aedes* mosquitoes. 

In contrast to mosquitoes, reproduction is seen to be highly divergent in tsetse flies. The analysis of the genome for these vectors reveals a reduction in the YPP gene cluster, *YP1*, as compared to closely related *Drosophila* flies, which have three or more YPPs in their genomes [61] (Table 1). On the contrary, though, a series of gene duplication events have been thought to result in the evolution of milk protein genes in tsetse flies. The milk gland is highly specialized in tsetse flies and secretes a complex mixture of stored lipids and milk proteins, providing nutrition and bacteria to the developing larvae. Thus, these genes required for lactation are highly evolved in these dipterans, in contrast to their counterparts in other insects [61]. Further studies comparing different components of reproductive biology targeting the steroid hormones or SFPs in tsetse flies are necessary (Table 1). No reproductive pathway component analysis has been conducted for *R. prolixus* or related species.

Unfortunately, this gap in information is due to the fact that the majority of omics-based investigations in hematophagous arthropods is disproportionately skewed toward Culicidae, particularly genera such as *Anopheles, Aedes*, and *Culex*. This is primarily driven by the epidemiological relevance of mosquitoes as vectors for pathogenic agents responsible for dangerous diseases like malaria, dengue fever, and Zika virus infections that drive the research and methods to control the disease spread. The availability of well-annotated genomic sequences for these genera further facilitates high-throughput omics analyses and tools like those seen in Pagete et al. and Brown et al. [84,85].

However, the analysis of lesser-known vector species could lead to the development of additional control techniques. The Mediterranean fruit fly, also known as Medfly (*Ceratitis capitata*), for example, is considered one of the worst agricultural pests in the world [86]. They serve as a fascinating case study for dynamic sperm storage and sperm selectivity, reflecting complex evolutionary strategies. The study by Scolari et al. [87] highlights that medfly females do not utilize their full sperm storage capacity post-mating. Instead, they allow sperm from multiple males to occupy different regions within the fertilization chamber, a strategy believed to optimize reproductive success and maintain genetic variability. This dynamic sperm storage behavior influences the paternal contributions to the offspring, thereby enriching the genetic pool. Interestingly, this nuanced approach to sperm storage mirrors similar behaviors observed in many blood feeding mosquito species. In both cases, females strategically allow sperm from multiple partners to co-exist in separate regions within their fertilization chambers. In the mating process of vertebrate blood meal-dependent mosquitoes, the MAGs produce secretions that serve as a medium for sperm movement and assist in their migration to the female’s spermathecae for storage [88]. Similarly, *Rhodnius prolixus*, also known as the "kissing bug", is a blood-feeding insect from the Reduviidae family. This bug is a main carrier for the protozoan parasite *Trypanosoma cruzi*, the cause of Chagas disease, which is a significant public health concern in Latin America [89]. In the context of blood-feeding insects like *Rhodnius prolixus* and mosquitoes, these nutrients often originate from host-derived blood meals. The blood-feeding behavior not only provides the resources for oocyte and egg development but also acts as a transmission route for disease agents, such as *Trypanosoma cruzi* in *Rhodnius prolixus* and various pathogens like the Plasmodium parasite and the Dengue virus in mosquitoes [75,90]. Thus, studying the reproductive biology of these vectors through an omics approach is essential not just for the understanding of their lifecycle but also for its implications in disease transmission and vector control.

Information obtained from these aforementioned evolutionary studies would provide us with strategies for targeting these complex physiological processes. If successful, the implementation of these strategies could lead to the prevention of reproduction in several insect vectors, which may eventually lead to a reduction in the transmission of vector-borne diseases globally. By pinpointing specific genes and molecular pathways, these vulnerabilities could be harnessed to design novel molecular or genetic interventions. One such approach could involve genetic engineering, where interference with key genes in the reproductive pathway of a vector like *Ae. aegypti* could yield significant effects on its reproductive behavior. These interferences could lead to chain reactions that would potentially disrupt vector mating patterns or reduce the viability of offspring. However, these genomic studies are only one preliminary component of vector control, as further approaches can be developed by delving into vector species from a transcriptomic perspective. The approaches developed from a transcriptomic analysis could lead to a comprehensive understanding of the regulatory reproductive mechanisms governing the reproduction of each species.

## 3. Transcriptomic Studies

During mating, most of the male insects release their spermatozoa into the female reproductive tract. These spermatozoa are accompanied by SFPs [54,91]. SFPs are synthesized in the MAGs [45]. Male insects are known to deplete their seminal fluids during repeated insemination and eventually need to be replenished, which can be achieved in about 48 h in *Ae. aegypti* males when allowed to recover post-mating [92]. Short-read RNA sequencing of MAGs and testes revealed that most upregulated transcripts encode for the highly abundant SFPs or sperm proteins [93] (Table 1). Based on the enrichment of male-biased genes, it was evident that genes encoding for SFPs were enriched on chromosome 1, which also has the region of male sex determination *Nix* [94] (Table 1). Interestingly, though, sperm proteins were enriched on chromosome 2, which raises questions regarding the functional significance of this pattern and needs further experimentation. Going forward, it would also be interesting to conduct a functional analysis of all of the identified seminal proteins in these mosquito species along with their characterization in other disease vectors. It would also be interesting to conduct knockdown or ectopic expression studies for candidate proteins to verify their roles in post-mating female behavior, which could eventually be used to develop new vector control targets.

Post-mating, females have an induction of transcriptional response, which brings about many changes that could influence the female’s ability to transmit the disease or affect mosquito reproduction. These changes are most likely attributed to the transfer of seminal fluids donated by the males, as discussed in the previous section. Analysis of the recently published transcriptomes of *An. gambiae* [78,95] and *Ae. aegypti* [96] reveal interesting transcriptional patterns that occur in these female vectors post-mating. As described earlier, 20E transfer reduces the chances of re-mating in females and induction of host-seeking for vitellogenesis. Apart from contributing to monandry, 20E also induces the *HPX15* gene to produce heme peroxidase 15 in sperm storage organs, maintaining fertility post-mating [97] (Figure 1; Table 1). Furthermore, genes involved in regulating energy metabolism, cellular transport, and oxidative stress pathways were also seen to be most differentially regulated in the spermatheca post-mating [97].

Previously, microarray studies were conducted to identify post-mating transcriptional response in *An. gambiae* 2, 6 and 24 h post-mating [78]. The study was conducted on whole mosquitoes, lower reproductive tract (LRT), and gut tissues. Changes in the expression of some of the candidate genes were also seen to be retained up to 4 days post-mating, suggesting permanent changes in the females post-mating. These transcriptional changes were also associated with a structural change in the female atrium [78]. Thus, it was considered that the female response to mating is characterized by the transcriptional changes in the atrium and spermatheca within 24 h post-mating [78]. However, when the role of the sperm in inducing the transcriptional changes was assessed, it was observed that mating with sperm-less males also induced the same post-mating response in *An. gambiae* [78], leaving a knowledge gap about the molecular basis of the male-induced gene response affecting the reproductive biology of *An. gambiae*. Further, in a study conducted by Thailayil et al. [95] that used wild-caught *An. gambiae* mosquitoes from Burkina Faso, 10 genes that were differentially (either up or down) regulated were selected from Rogers et al. [78] to assess their expression in the lower reproductive tract of the female *An. gambiae* mosquitoes 1 and 4 days post-mating. From these studies, only one out of the 10 genes identified, mating induced stimulator of oogenesis (*MISO*), has been associated with induction of oogenesis post blood feeding, via 20E [98] (Figure 1; Table 1). The functions of the remaining nine genes concerning mosquito reproductive biology have not yet been determined. In the same study, the authors also identified seven factors differentially expressed in the female carcasses that could be related to reproductive success or mating behavior in female mosquitoes [78,95].

Large-scale transcriptomic response screening has only been conducted using the Illumina platform in *Ae. aegypti* to identify the gene response in females 0, 6, and 24 h post-mating [99]. Most of the transcripts that were differentially expressed at 0 h post-mating was also found to be differentially expressed at 6 and 24 h at similar levels as compared to virgin females. Interestingly, this screening mapped 60 extra transcripts at 0 h post-mating as compared to the virgin females, which is suggestive of these being transferred from males. However, only 20 genes were identified to be coding for male seminal fluid protein genes [99]. The differentially expressed genes at the different time points were associated not only with promoting blood feeding or oviposition but also with immunity, which may have implications concerning survival and vector competence.

The other insect vectors discussed in this review, however, seem to be understudied in correlation with post-mating gene response in females. The transcriptomic studies conducted in both tsetse flies and kissing bugs have been restricted to the elucidation of the transcriptional changes in reproductive tissues of the female flies that affect fecundity outcomes [70,100,101,102,103]. The role of tsetse flies in the transmission of sleeping sickness has been known for more than 100 years, with initial reports by Sir David Bruce [104]. Female tsetse flies remain receptive for the second insemination if it occurs immediately after the first [50], but slowly develop refractoriness between 24–48 h post-mating [69,70]. With initial hybridization studies, it has been known that female tsetse flies exhibit a post-mating barrier to the inheritance of genetic material, which is responsible for reduced fecundity [105,106] or maternal factors-induced male sterility [107]. It is not only the transfer of SFPs that induces monandry; it is also necessary that the products from MAGs are assembled to form a spermatophore in the female uterus post-mating [50] (Table 1). The presence of a spermatophore in female tsetse flies following mating may give rise to a physical barrier, potentially preventing the sperm from a second male from entering. However, a comprehensive understanding of spermatophore components and their roles during fertilization or in the post-mating response of female tsetse flies remains elusive. In contrast, kissing bugs demonstrate distinct reproductive behavior when compared to other vectors. These females are recognized as polyandrous, readily engaging in multiple matings throughout their lifespan, often attracting different males through the secretion of sex pheromones [71]. However, the roles of specific genes responsible for these phenomena remain unstudied. Overall, mating and blood meal-induced oviposition are complicated processes occurring in most of the insect disease vectors, yet very few genetic screenings have been conducted to address the post-mating response in females. Comprehensive understanding of the genetic basis of reproductive biology in female insects is still lacking, which in turn is consequentially a hindrance in the development of new vector control strategies.

Though there is potential in these transcriptomic discoveries, there still remains a significant gap in the comprehension of the complexity of the genetic information that affects the reproductive biology of these insect vectors. The transcriptomic approach offers advantages by considering the dynamic nature of genetic information storage in each species. This, in turn, has significant implications for the development of new vector control strategies, making it challenging to grasp the intricacy of transcribed data without incorporating proteomic analysis. Proteomics, a comprehensive large-scale study, aims to identify proteins and interpret their composition and structure. The proteome, originating from the genome housing an organism’s complete genetic blueprint, encompasses the entire collection of proteins within an organism’s cells at a specific timepoint. Unlike the genome, which remains constant throughout an organism’s life, the proteome exhibits temporal variability and can differ among individual cells. By understanding the proteome, we can hope to understand not only the key components of protein synthesis in the reproductive cycle of the vectors but also elucidate the importance of protein–protein interactions and their effects on the insects in question.

Although the transcriptomic studies would shed light on the gene regulatory networks that facilitate the production of these vital proteins, one potential technique developed by proteomic analyses could emerge from a comprehensive analysis of male vector seminal fluids. The collected information could lead to the identification of novel proteins and pathways, elucidating their roles in modulating post-mating responses in female vectors. Thus, utilizing proteomic analyses to fathom this reproductive complexity holds promise for the development of vector control efforts.

## 4. Proteomic Studies

Proteomics is a multidimensional field that goes beyond merely studying protein synthesis. It delves into the critical interactions between proteins and examines how these interactions influence the biological functions and behaviors of the organism, including their effects on fertility in various blood-feeding insect species. The results of protein–protein interactions can be observed by studying their effects on fertility in different blood-feeding insect species. Many of the reproductive proteins produced by male insects can be responsible for changes in female behavior after mating has taken place. During mating, most of the male insects release their spermatozoa into the female reproductive tract (FRT). These spermatozoa are accompanied by SFPs [54,91], and various studies have shown that SFPs interact with the proteins found within the FRT, resulting in fertilization and reproductive success, which are caused by a variety of post-mating responses [108]. After transfer into the female reproductive tract, MAG extracts or SFPs modulate several physiological and behavioral changes in the females which include, reduction in mating receptivity, induction of oogenesis and oviposition, increased host-seeking/blood intake, changes in sleeping patterns, and increased immune responses [53,82,109,110,111,112,113,114,115,116]. These post-mating changes are not simply restricted to behavior, as studies have shown that physiological and morphological changes also take place [65]. These changes can affect the female mosquitos’ receptivity to mating, as it is believed that the purpose of these induced changes is to ensure a successful copulation [57]. Most female mosquitoes are thought to only mate once during their lifetime but can produce eggs during each gonotrophic cycle post-blood meal. Re-mating is a common phenomenon known to occur in male mosquitoes but rarely observed in females [63]. In *Ae. aegypti* [64] or *An. gambiae* [65], the refractoriness to multiple copulations is largely a result of male SFPs that are transferred during mating. In *An. gambiae*, impaired sexual transfer of 20-hydroxyecdysone (20E) has been shown to reduce monandry, making it an essential post-mating response in these female mosquitoes [65].

After the gonotrophic cycle has occurred and fertilization has begun, the FRT will change to accommodate the fertilized eggs and allow their maturation [65]. The proteins found in the male’s seminal fluid initiate these changes only after their introduction to the female’s reproductive tract, as studies have also shown that the proteome of virgin females will differ from the proteome of mated females [108]. Apart from biochemical or genetic changes that are introduced in females post-mating, it has been observed that the FRT undergoes structural changes in the smooth and rough endoplasmic reticula, which disintegrate and cause the mitochondria to disperse throughout the atrium cells that surround the smooth endoplasmic reticulum in virgin *An. gambiae* females [78]. These structural changes are believed to contribute to the restricted re-mating in female insects [78]. The seminal fluids in arthropods are known to contain some proteases/protease inhibitors, lectins, prohormones, peptides, and protective proteins such as antioxidants [54,91,115]. Studies in crickets and *An. gambiae* have revealed that the seminal fluid may also contain non-proteinaceous components, such as prostaglandins [117,118] or steroid hormone 20-hydroxyecdysone [119], respectively. However, studies have only focused on the effects associated with the transfer of SFPs. Also, functional characterization of individual components from the SFP complex from male disease vectors remains elusive. It has only been over the past few decades that large-scale mass spectrometry-based proteomic analyses have been conducted in *An. gambiae* and *Ae. aegypti* [57,78,93,120,121] to elucidate the protein architecture of SFPs.

The FRT has been studied across various types of taxa, and the protein composition of its tissues and fluids has shown variations across different points in the timespan of female development. It is hypothesized that the fluid secretions of the reproductive tract are necessary to allow for the molecular interactions that cause the SFP modifications that in turn lead to fertilization [122]. The protein–protein interactions between sperm and the FRT are noteworthy not only due to the changes they induce in female behavior but also for their effects on female survival. While the MAG proteins found in seminal fluid have been shown to decrease the lifespan of *D. melanogaster* [57], one study has shown that *Ae. aegypti* females have demonstrated an increased survival after copulation [93] (Table 1). This increase in female survival could be due to the transfer of the SFP molecules that occurs with insemination. The different mating effects on the lifespan of *Drosophila* as compared to *Ae. aegypti* stem from the variations in male ejaculate composition across species. The specific protein or molecular interaction responsible for the increased survival of the *Ae. aegypti* females has not been identified, and further studies are necessary to understand this effect. However, the study by Sirot et al. [121] identified 22 putative mRGPs and 93 SFPs along with 52 sperm proteins that are transferred to *Ae. aegypti* females post-mating (Table 1). The alignment of the identified SFPs to *Drosophila* orthologs suggested that these proteins may play a role in the utilization of sperm or ecdysteroidogenesis [121]. Following these initial studies, Degner et al. [93] used an isotopic labeling approach to isolate 870 and 280 male-derived sperm proteins and SFPs, respectively. These proteins were identified to be associated with male reproductive biology in *Drosophila* [123,124] and *Ae. albopictus* [115]. Moving ahead, it would also be interesting to conduct a functional analysis of all of the identified seminal proteins in these mosquito species along with their characterization in other disease vectors. It would also be interesting to conduct knockdown or ectopic expression studies for candidate proteins to verify their roles in post-mating female behavior, which could eventually be used to develop new vector control targets.

The contrasting outcomes from the protein–protein interactions of *Drosophila* and *Ae. aegypti* or *Drosophila* and *An. gambiae* are not the only examples of alternative effects in reproductive biology across different insect species. The tsetse fly *Glossina fuscipes* is another type of blood-feeding insect species demonstrating proteomic differences when compared to the common fruit fly. Tsetse fly reproduction occurs in a manner similar to that of *Ae. aegypti*, when the *Glossina* males transfer seminal fluid containing MAG proteins to the female reproductive organs (Table 1). There is only one study so far, to the best of our knowledge, that identifies the spermatophore-associated proteins in tsetse flies [50]. This study in *Gl. morsitans* revealed a small number of proteins secreted by male MAGs or testes; however, the seminal proteins showed low sequence similarity with *Drosophila* or *An. gambiae*, making them highly evolving genes that could be species-specific [50] (Table 1).

The FRT receives a large amount of foreign material, including microbes. Infectious agents affect the proteomes of other blood-feeding insects. Fecundity is seen to decrease in *Gl. fuscipes* females that are infected with the endosymbiotic bacterium *Spiroplasma* after taking a blood meal. The bacterium is capable of affecting the reproductive physiology of its insect hosts, and *Drosophila* females also demonstrated physiological changes after being infected with the pathogen [125]. In *Drosophila*, the short wing protein ‘*sw*’ is affected when considering the reproductive biology of these insects. One study suggests that the *Gl. fuscipes* ortholog of the ‘*sw*’ protein found in *Drosophila* could be responsible for the reduction in fitness of the insects after a *Spiroplasma* infection [126]. The impact of the bacterium on the ‘*sw*’ ortholog remains unclear, and as such, several experiments need to be conducted to determine how this information can be useful when attempting to implement successful methods of vector control. Interestingly, to protect against newly introduced microbes during mating in the reproductive tract of females, SFPs have been characterized to exhibit antibacterial properties in *Drosophila* [55,127]. Mated kissing bug females require multiple blood meals for their nutritional development as well as egg laying. It is hypothesized that stimulations responsible for egg formation occur in the ovaries and result from hormonal interactions with the juvenile hormone and ovarian tissues [128]. One study has shown that *T. cruzi* infection affects the trypsin inhibitor *RpTI* and increases the expression of the gene coding for the aspartyl protease cathepsin D [129]. This same study asserts that while its results are preliminary, further proteomic analysis of *R. plexus* can lead to successful control strategies for Chagas disease. As such, it is not excessive to assume that significant proteomic effects are taking place in the reproductive system of these kissing bugs as in their digestive system.

The control methods mentioned in this section are only preliminary examples of the benefits from developed vector control strategies. Within this article, we have presented multiple examples for the control methods developed from these omics-based studies. Each study, whether it be genomic, transcriptomic, or proteomic, offers its own set of potential control techniques and benefits. Once enough research studies are conducted, we can potentially develop the appropriate control approach tailored to specific vector species and integrate additional adaptations to best suit the particular species under consideration. However, it is worth noting that we have failed to explore the potential avenues that could emerge from metabolomic studies. Metabolomics encompasses the understanding of the metabolic processes within each organism, offering valuable insight into the networking of biochemical compounds of each organism. These studies could also provide a more holistic understanding of biological systems of vector species and hold promise for revealing novel control strategies.

## 5. Metabolomic Studies

Metabolomics considers metabolites and aims to categorize and identify their involvement within various chemical processes. This systematic study provides characteristic fingerprints that specific cellular processes yield by studying their small-molecule metabolite profile [130,131]. Different techniques like spectroscopic (nuclear magnetic resonance), spectrometric (mass spectroscopic), and separation techniques (LC, GC, supercritical fluid chromatography, CE) are used depending on the sample and the properties of the metabolites [132]. Many metabolites are critical for the reproductive capacity of blood-feeding female insects and their male counterparts. Understanding their roles and regulatory molecular mechanisms can help elucidate new perspectives to control insect populations [133].

The fat body in insects plays a significant role in immunity, synthesis of hemolymph substances, and reproduction [134,135]. A study by Price et al. involved the metabolomic analysis of female adult *Ae. aegypti* mosquitos’ fat bodies concerning the larval nutritional regimens [136]. Nutritional restriction enabled the analysis of mosquito groups, revealing a large number of significantly differentially expressed genes. The metabolomic analysis showed that lysine, tryptophan, aspartic acid, histidine, isoleucine, threonine, and alanine were found in significantly increased concentrations in small and large mosquitoes. It was also observed that the energy molecules like ATP and NADH, which are derived from blood meals, were used to replenish the female mosquitos’ reserve in smaller mosquitos rather than being used for reproduction [136]. A similar study by Horvath et al. showed how smaller females prioritize the degradation of proteins from the first blood meal and reserve them to produce energy, whereas in the case of large females, the nutrients from the first blood feed are used to support vitellogenesis and egg production [137].

Insulin-like peptides (ILPs) play essential roles in metabolic homeostasis and reproduction in mosquitoes, mainly *Ae. aegypti*. Brown M et al. showed how endogenous ILPs expressed in the brains of female mosquitoes activate insulin-signaling pathways in the ovaries that drive egg maturation [138] (Table 1). Pietri et al. performed a metabolomic analysis in the female *Anopheles stephensi* mosquito which revealed that egg maturation is stimulated by the activation of the insulin signaling pathway in the ovaries by an endogenous ILP in the brain [139]. Further, a lipidomic study was conducted from the fat bodies of female adult *Ae. aegypti* mosquitoes, measuring the lipids in the pre- and post-blood meal periods spanning the entire vitellogenic cycle of these mosquitoes (Figure 1). Four hundred fifty-six lipids were identified and divided into six groups. The most significant shifts in the lipid components were observed in two stages of PBM: the first change right after the blood meal (0 h PBM) and the second change between 16-30 h PBM. The second stage also showed a substantial increase in the cell membrane lipids, signifying the development of embryos during the reproductive process. About 392 lipids in the 16-30 h PBM duration increased significantly and were expected to contribute to mosquito egg development [140]. A study by Rivera-Perez et al. showed the metabolites in the female *Ae. aegypti* mosquito’s four nutritionally different corpora allata (CA; a pair of endocrine glands in the mosquito), observing the crucial mevalonate pathway (MVAP), which synthesizes juvenile hormone (JH) and additional metabolites [141] (Table 1). The MVAP is responsible for many biological functions and the supply of isoprenoid precursors that control JH synthesis. JH in the previtellogenic phases of the female *Aedes aegypti* mosquito controls the nutrients provided to the ovaries and impacts the fate of vitellogenic follicle development after a blood meal [141].

Huck et al. in 2021 studied the MAGs in a male Northern house mosquito, *Culex pipiens* [142]. The adult males were divided among three different dietary treatments: 3, 10, and 20% sucrose diet, respectively. NMR spectroscopy-based metabolic composition showed differences like metabolites in mosquitoes raised on different diets. The nutritionally depleted males (raised on 3% sucrose) showed reduced formic acid levels and increased glucose and lactic acid levels. Proteins and lipid composition correlated positively with the sugar in the diet. The males fed on 3% sucrose significantly affected the reproduction by lowering the hatching rate of the progeny, although no significant effect was observed in the eggs laid [142]. Another metabolic component analysis indicated substantial changes in the levels of amino acids and carbohydrates in *C. pipiens* dehydration, affecting reproduction negatively [143].

Another study was conducted to observe the sporogony of *P. falciparum* in *Anopheles* sp., where miR-276 was targeted. The anabolism to catabolism switch in the insect fat body is enabled after blood feeding in the insect to produce a cascade leading to the release of glycogen and other lipid reserves [144,145]. Among the negative regulators of this, miR-276 drives a shift in metabolism in the fat body where the catabolic amino acid metabolism switches to anabolic to reset the process. Metabolites from the miR-276 knockdown mosquitoes were detected, showing active restriction of the reproductive investment by the mosquito and driving the *P. falciparum* sporogony. In the miR-276 depleted cohort, the metabolomic analysis showed increased fecundity and average larvae hatching rates [145] (Figure 1).

Stearoyl-CoA desaturase (SCD) plays a critical role in the digestion of the blood meal in female *Anopheles* mosquitoes [146] (Figure 1; Table 1). It is crucial in the conversion of saturated fatty acids to unsaturated fatty acids, which play an important role in maintaining cell membrane fluidity. Metabolic profiling in *SCD1* knockdown *Anopheles coluzzii* mosquitos revealed a depletion of lipid droplets that led to a detrimental effect on oocyte maturation, affecting their reproductivity [146].

The microbiome is also seen to affect the metabolism and reproductive capacity of these blood-feeding insects. For example, in *Gl. morsitans* tsetse flies, the absence of the symbiotic *Wigglesworthia* disrupts carbohydrate and amino acid metabolism [147] and depletes the host of *Wigglesworthia-*derived vitamin B [148], which directly impacts the metabolism, physiology, and reproduction of the fly.

Similarly to the additional omics studies mentioned throughout our article, metabolomic studies hold significant promise in the realm of species vector control. By scrutinizing unique components of the small-molecule metabolites found within the reproductive pathways of these insect species, these studies offer unparalleled insights into insect population management. The knowledge acquired from additional metabolomic research ventures can lead to the development of targeted interventions, including novel insecticides or repellents that disrupt essential metabolic processes. Moreover, understanding the metabolic interactions between vectors and the pathogens they carry can allow scientists to postulate or develop strategies leading to the blockage of transmission cycles. Metabolomics also aids in assessing the impact of environmental factors and microbiomes on vector metabolism, providing a holistic understanding of vector biology and ultimately helping curb the notably adverse and fatal disease outbreaks that are currently impacting multiple countries and the unfortunate individuals afflicted by them. In essence, metabolomic studies offer a promising avenue for more effective and sustainable vector control measures, with the potential to significantly reduce the burden of vector-borne diseases on global public health.

## 6. Conclusions

In insects, the reproductive systems of both males and females are highly complex and have evolved to facilitate the reproductive success of individuals. Through the elucidation of in-depth genome sequencing, novel gene characterization has helped us understand the sequence variability and evolution of genes compared to known homologs. Genes such as *AgTG3*, *Or, yellow*, *HP1*, and *YP1* have undergone significant evolution in these insect vectors, contributing to increased propagation and survivability. Understanding the genomic structure of these insects allows for potential biomarker targets that can help in the reduction of vector propagation. Transfer of the male seminal fluid is a common phenomenon that is observed in the sexually reproducing taxa. The molecular diversity of this fluid in particular insect species defines their divergence and provides species-specific reproductive targets that may be used for potential vector control strategies. Transcriptional changes such as the expression of *HPX15* have been shown to help in maintaining fertility post-mating. Seminal fluid factors can alter the transcriptome of females and cause physiological changes in immunity and changes in atrium and spermatheca as well as behavioral changes such as blood feeding and oviposition, implicating their importance in survivability. However, there is a need for elucidation of these transcriptional factors and gene expressions, such as those from *An. gambiae.* Additionally, it is worth noting that the introduction of SFPs has profound effects on their proteome regarding female behavior and physiology. SFPs can alter the reduction in mating receptivity and induction of oogenesis and oviposition, increase host-seeking/blood intake, induce changes in sleeping patterns, and increase immune responses. The effects of these protein–protein interactions and the effects that the microbiome has on these proteins, such as the short-wing protein, can provide key insights into target proteins that could be used to reduce vector feeding and mating. Among recent studies on the reproductive biology of *Anopheles gambiae* mosquitoes, a study focused on the role of male-transferred steroid hormones in influencing female reproductive success. Notably, males produce significant amounts of the steroid hormone 20-hydroxyecdysone (20E), primarily in the accessory glands, and transfer it to females during copulation. This transferred 20E interacts synergistically with a female-specific protein, MISO, to regulate oogenesis in a mating-dependent manner. Additionally, males transfer Juvenile Hormone III (JH III), which appears to modulate female resource allocation toward reproductive ends, including ovarian lipid storage and previtellogenic resorption rates. Interestingly, the amount of JH III transferred is contingent upon the male’s nutritional status [98,119,149] (Figure 1; Table 1). Another seminal work on *Rhodnius prolixus* illuminated the dearth of genetic understanding in triatomine male reproductive biology, a field largely overlooked. The study provided a comprehensive testicular transcriptome, unveiling a complex landscape where approximately 50% of transcripts remain unknown. Intriguingly, genes traditionally associated with hematophagy—such as lipocalins, serpins, and lysozymes—display high abundance in testicular transcripts, raising questions about their evolutionary co-option and unknown roles in male fertility. This study serves as a critical foundation for future investigations into triatomine male biology and offers potential targets for disrupting vector fertility [150]. Similarly, another comprehensive study delved into the transcriptomic changes in the central nervous system, fat body, and ovaries of *Rhodnius prolixus* post-blood meal, elucidating their significance in egg production and, consequently, vector proliferation. The authors adeptly bridged the tissue-specific responses with overarching themes in lipid, protein, and trehalose metabolism, neuroendocrine signaling, and immune system interactions [128]. Insulin-like peptides activate insulin-signaling pathways in the ovaries that drive egg maturation. Lipid class composition and changes in females are crucial for the development of embryos. MVAP is responsible for juvenile hormone synthesis and affects the vitellogenic follicle development in *Aedes aegypti*. Stearoyl-CoA desaturase is crucial for digestion after blood feeding. The metabolomic composition of these insects can further elucidate factors that contribute to behavior, mating, and blood feeding as well as fill in knowledge gaps when analyzing insect mechanisms through other omics approaches individually. Efforts need to be put into species sampling and improving the omics knowledge base to characterize insects and the impact that SFPs have on females’ post-mating responses in blood-feeding disease vectors. Genomic, transcriptomic, proteomic, and metabolic analyses are key to the exploration of insect physiology and behavior. A combination of these omics approaches can help improve our understanding of ways to control vector populations in disease-endemic areas where the burden of the illnesses transmitted is most prominent, thus reducing the global burden of vector-borne diseases.

The multidimensional understanding of the studies presented in this article leverages the positive insights derived from these analyses, allowing scientists to identify the key vulnerabilities in these vectors, unveiling potential targets for novel control strategies. These may include the development of precise and environmentally sustainable interventions, such as genetically modified insects, targeted insecticides, or strategies to disrupt key metabolic pathways essential for vector reproduction and disease transmission. Ultimately, the integration of the presently collected omics data offers a promising opportunity to advance our understanding of vector biology and develop more effective and sustainable approaches to control vector populations, reducing the impact of vector-borne diseases and improving public health worldwide.

## Figures and Tables

**Figure 1 insects-14-00802-f001:**
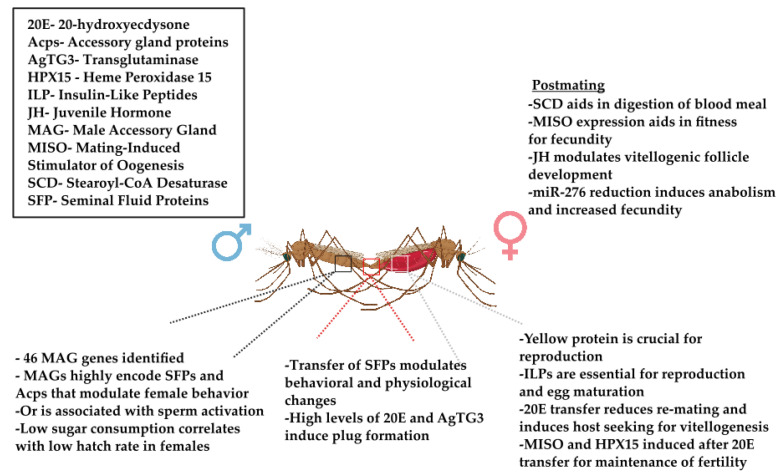
The molecular dynamics that influence behavior and physiology in the species *Anopheles*. Here, we see both male and female factors and their responses. The top left box shows the acronyms of the modulating factors.

**Table 1 insects-14-00802-t001:** A table representing the modulating factors of various insects. The table is divided into four omics categories, followed by species that were studied in the omics category and the functionality of the molecular target. Please note some species have overlapping targets.

Omics Analysis	Organism	Target	Location of Target Analysis	Function/Outcome
Genomics	*Glossina morsitans*	Yolk protein (yp2)	Ovaries	Expressed in the ovaries and involved in oogenesis
	*Aedes aegypti*	Male reproductive gland protein-encoding genes	Male accessory glands	Multiple genes encoding the respective accessory gland proteins, influence female behavior, physiology, survival and reproduction
		Heterochromatin Protein 1 (HP1)	Seminal fluids	Induce female refractoriness to mating
		M-locus gene (Nix)	Chromosome 1	Male sex locus determination gene
	*Anopheles gambiae*	Male accessory gland genes (MAGs)	Male accessory glands	Multiple genes that encode for the respective accessory gland proteins
		Yellow, -b, -e, -g	Ovaries	Crucial for female reproduction
		Transglutaminase (AgTG3)	Seminal fluids	MAG-specific enzyme involved in formation of mating plug
		Odorant receptor (Or)	Antennae/Palps	Sperm activation
	*Anopheles stephensi*	Yellow, -b, -e, -g	Ovaries	Consistent with roles in female reproduction
	*Stomoxys calcitrans*	Odorant receptor (Or)	Maxillary palps	Potential role in sperm activation
Transcripts	*Anopheles gambiae*	Mating induced stimulator of oogenesis (MISO)	Atrium	Induction of oogenesis post blood feeding
		Heme peroxidase 15 (HPX15)	Sperm storage organs	Involved in activation by sex that are important to preserve the functionality of stored sperm and long-term fertility
Proteomic	*Anopheles gambiae*	Accessory gland genes (Acps)	Accessory gland	Control post-mating behavioral changes in females
		Seminal fluid proteins(SFPs) / Male accessory gland proteins (MAGs)	Male accessory glands	Refractoriness to multiple coapulations
	*Aedes aegypti*	Seminal fluid proteins(SFPs) / Male acessory gland proteins (MAGs)	Male accessory glands	Refractoriness to multiple copulations and/or ecdysteroidogenesis
		Putative male reproductive gland proteins (mRGPs)	Seminal fluids	Induces behavioral and physiological changes in females
	*Glossina morsitans morsitans*	Seminal fluid proteins(SFPs) / Male acessory gland proteins (MAGs)	Male accessory glands	Modulate female reproductive physiology and behavior, impacting sperm storage/use, ovulation, oviposition, and mating receptivity
	*Glossina fuscipes*	Ortholog of short wing proteins (sw)	Male testes, ovaries	Possible reduction in fitness of the insects after a Spiroplasma infection
Metabolomic	*Anopheles gambiae*	20-hydroxyecdysone (20E)	Seminal fluids/ Accessory glands	Formation of mating plug
	*Anopheles coluzzii*	Stearoyl-CoA desaturase (SCD)	Midgut lumen	Digestion of the bloodmeal in females, oocyte maturation, maintains cell membrane fluidity
	*Anopheles stephensi*	Insulin-like peptides (ILPs)	Brain	Stimulates egg maturation
	*Aedes aegypti*	Isoprenoid precursors for mevalonate pathway (MVAP)	Corpora allata	Controls synthesis of important metabolites most notably juvenile hormone
		Juvenile Hormone (JH)	Corpora allata	Controls nutrients provided to ovaries, impacts the fate of vitellogenic follicle development
		Insulin-like peptides (ILPs)	Brain	Metabolic homeostasis and reproduction

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
