# Peer review of "Current Status of Omics Studies Elucidating the Features of Reproductive Biology in Blood-Feeding Insects"

_insects, 2023, doi:10.3390/insects14100802_

Round 1
Reviewer 1 Report
Current Status of Omics Studies Elucidating the features of reproductive biology in blood-feeding insects
This review is generally shallow and lacking. The levels of the writing between the different sections are not even. This review requires significant changes to be considered for publication.
Major comments:
1) Sections 1 and 2 are poorly written. There is a considerable difference between them and the other sections in the manuscript.
2) Section 1 is the introduction of the review. It should explain to the naïve reader in detail the importance of the review and the epidemiological significance of the pathogen transmitted by blood-feeding insects. It should explain the biology behind reproduction and oogenesis and that even though only females transmit the pathogen, male biology is of importance; why is it of importance and that this review explores knowledge enquired on both males and females to find new strategies to reduce the burden of the diseases transmitted by the insects and improve public health. Moreover, the knowledge available from omics studies of blood-feeding insects is mainly based on mosquitos. This should be explained here.
3) The authors forget to elaborate in the introduction on the importance of yellow fever and Zika, which brought a lot of money and attention to this neglected field. The authors also forgot to mention sand flies.
4) Section 2 should follow the writing logical order and quality of section 3: a short explanation of the biological function in the section and its importance and then a review of the current knowledge. Lines 131-134 are the lines that should conclude the introdction, not this section.
5) The last paragraphs in all sections are entirely detached from the unit—no flow, no logical context, etc.
6) Section 4: It is known from other Diptera that females mate only once but with several males. The females then store the sperm from each male in a different spermatheca and use the best quality sperm first (look for studies on the Mediterranean fruit fly). What is known about this in mosquitos? In the context of the section, one should speculate, address and discuss that different sperm proteins may be excreted in different levels from different males, how this may affect the female and the cumulative effect of the sperm even if there is no current knowledge on the subject.
7) Lines 330-343 should be part of the introduction.
8) Lines 423-431 are about ticks. Why ticks? They are blood-feeding arthropods, but they are not insects! Either explain adequately or take it out. It does not belong in this review.
9) Omics and bioinformatics are not enough. After analyzing all omics and identifying the hypothesis they lead to, Behavioral studies must be conducted to complete the research and draw the correct conclusions that will lead us to the best strategies to fight blood-feeding insects. This must be addressed in the conclusion paragraph.
Author Response
Major comments:
1) Sections 1 and 2 are poorly written. There is a considerable difference between them and the other sections in the manuscript.
We thank the reviewer for this thoughtful comment. Sections 1 and 2 have been rewritten to include all of the information that the reviewers suggested. It has been completely revamped and contains more information and a better flow similar to that in the other sections.
2a) Section 1 is the introduction of the review. It should explain to the naïve reader in detail the importance of the review and the epidemiological significance of the pathogen transmitted by blood-feeding insects.
We agree with the reviewer and we have added additional content detailing the importance of the review and the epidemiological significance of the pathogen transmitted by blood-feeding insects that is relevant to the epidemiological significance which can be found in lines 34-103.
2b) It should explain the biology behind reproduction and oogenesis and that even though only females transmit the pathogen, male biology is of importance; why is it of importance and that this review explores knowledge enquired on both males and females to find new strategies to reduce the burden of the diseases transmitted by the insects and improve public health.
As per the reviewer’s suggestion, we have updated the review to mention Oogenesis and reproduction in multiple sections throughout the review. They can be found in sections 2-5. Each section mentions their significance respective to the omics analysis. Section 2 lines 160 to 182 introduces the concept of oogenesis and reproductive importance. The significance of Oogenesis and reproduction are tied to the genomic expression patterns.
2c) Moreover, the knowledge available from omics studies of blood-feeding insects is mainly based on mosquitos. This should be explained here.
We elaborate on this comment in lines 238-245 referring to a disproportionate skew towards Culicidae.
3a) The authors forget to elaborate in the introduction on the importance of yellow fever and Zika, which brought a lot of money and attention to this neglected field.
We thank the reviewer for bringing this to our attention. Lines 55 through 61 elaborate more on yellow fever and Zika, these are also briefly mentioned in lines 35-38.
3b) The authors also forgot to mention sand flies.
We thank the reviewers for this suggestion. Lines 82 through 86 elaborate a bit more on sand flies and their characteristics.
4) Section 2 should follow the writing logical order and quality of Section 3: a short explanation of the biological function in the section and its importance and then a review of the current knowledge.
Section 2 has been completely rewritten, from lines 118 to 198 and 238 to 286. The content follows a better introduction to the genomics theme as well as a more fluid transition.
4a) Lines 131-134 are the lines that should conclude the introduction, not this section.
These lines have been removed since the sections were redone.
5) The last paragraphs in all sections are entirely detached from the unit—no flow, no logical context, etc.
As per the suggestion of the reviewer, we have added transitions between sections for a better flow and logical context. Sections 2-5 all have new closing transition paragraphs.
6) Section 4: It is known from other Diptera that females mate only once but with several males. The females then store the sperm from each male in a different spermatheca and use the best quality sperm first (look for studies on the Mediterranean fruit fly). What is known about this in mosquitos? In the context of the section, one should speculate, address, and discuss that different sperm proteins may be excreted in different levels from different males, how this may affect the female, and the cumulative effect of the sperm even if there is no current knowledge on the subject.
We thank the reviewer for this suggestion. Lines 246-260 expand on these ideas and describe the Mediterranean fruit fly and its sperm storage mechanisms and then parallel this topic in mosquitos. We also elaborate on the sperm proteins and their effect on females. Lines 293 to 299 and 312-316 also elaborate more on sperm proteins and their significance as well as mentioning their importance throughout the article.
7) Lines 330-343 should be part of the introduction. (Removed)
As per the reviewer’s suggestion, we have removed this line.
8) Lines 423-431 are about ticks. Why ticks? They are blood-feeding arthropods, but they are not insects! Either explain adequately or take it out. It does not belong in this review.
We agree with the reviewer and lines 423-431 (previous version), as well as any mention of ticks have been removed from the article.
9) Omics and bioinformatics are not enough. After analyzing all omics and identifying the hypothesis they lead to, Behavioral studies must be conducted to complete the research and draw the correct conclusions that will lead us to the best strategies to fight blood-feeding insects. This must be addressed in the conclusion paragraph.
Per the reviewer’s suggestions, lines 652-678 have been added to elaborate further on metabolites and genes that affect insect behavior and targets that can be used to control these vectors. Lines 692-705 elaborate on the conclusion, which summarizes the previous points and gives an overall perspective on insect behavior and their reproductive intricacies.
Reviewer 2 Report
Comments:
In this review, the authors present the current knowledge about the genomic, transcriptomic, proteomic, and metabolomic studies describing the reproductive biology of four blood-feeding female insect vectors. They also compared the shared and species specific male-derived triggers that are transferred during mating to the females and the subsequent post-mating genetic responses. It is useful to help understand the reproductive components that are still unexplored in some insect vectors, and could help identify potential targets for vector control.
Typos:
1. Line 3, the dot should be deleted.
2. Line 30 and Line 32, the numbers of references are wrong.
3. Line 62, the number of reference should insert after “female insects [23]......”
4. Line 67-73, [27] should be [24], [24, 25] should be [25, 26]. Please check the references and corresponding the numbers in the manuscript.
5. Line 538-539, “28. Neafsey, D. E. et al. ...... Science (1979) 347, (2015). 538; 29. Nene, V. et al. ...... Science (1979) 316, 1718–1723 (2007)”. Please check the years of the two references.
6. Line 130, “ Tsetse flies” should be “ tsetse flies”.
7. Line 177 &179, “et. al.” should be “et al.”
8. The article has some mistakes in sentence formation and grammar, it is suggested to get it corrected and revised for better understanding at many places (for example 225-226 needs a change as it is grammatically not correct).
9. The sentence in line 242-248 is copy of the sentence in line 91-98 . Please change the statment.
10. Line 601 and 605, the format of reference 62 and 64 are different with other references. Please change it.
Moderate editing of English language required
Author Response
Comments:
In this review, the authors present the current knowledge about the genomic, transcriptomic, proteomic, and metabolomic studies describing the reproductive biology of four blood-feeding female insect vectors. They also compared the shared and species specific male-derived triggers that are transferred during mating to the females and the subsequent post-mating genetic responses. It is useful to help understand the reproductive components that are still unexplored in some insect vectors, and could help identify potential targets for vector control.
Typos:
- Line 3, the dot should be deleted.
The line has been removed.
- Line 30 and Line 32, the numbers of references are wrong.
Reference numbers have been corrected
- Line 62, the number of reference should insert after “female insects [23]......”
The citation has been moved
- Line 67-73, [27] should be [24], [24, 25] should be [25, 26].
Citations have been corrected
- Please check the references and corresponding the numbers in the manuscript.
We thank the reviewer for bringing this issue to our attention. References have been redone to ensure correct correspondence.
- Line 538-539, “28. Neafsey, D. E. et al. ...... Science (1979) 347, (2015). 538; 29. Nene, V. et al. ...... Science (1979) 316, 1718–1723 (2007)”. Please check the years of the two references.
References have been corrected. These are now references 67 and 68 respectively.
- Line 130, “ Tsetse flies” should be “ tsetse flies”.
This has been corrected in the revised manuscript.
- Line 177 &179, “et. al.” should be “et al.”
This has been corrected in the revised manuscript.
- The article has some mistakes in sentence formation and grammar, it is suggested to get it corrected and revised for better understanding at many places (for example 225-226 needs a change as it is grammatically not correct).
We thank the reviewer for pointing this out. The article has been revised and edited to correct all sentence formations and grammatical issues.
- The sentence in lines 242-248 is a copy of the sentence in lines 91-98. Please change the statement.
We thank the reviewer for bringing this to our attention, the statement has been revised and changed along with the entire section.
- In lines 601 and 605, the format of references 62 and 64 are different from other references. Please change it.
We thank the reviewer for bringing this to our attention, the formatting issues have been corrected.